# Evaluation of Everyday Living Areas for Deinstitutionalized Community-Living People with Mental Illness

**Yuri Nakai and Hisao Nakai \***

Faculty of Nursing, University of Kochi, 2751-1 Ike, Kochi 781-8515, Japan; nakai_yuri@cc.u-kochi.ac.jp
**\*** Correspondence: nakai_hisao@cc.u-kochi.ac.jp

**Abstract:** Deinstitutionalization of psychiatric care has been associated with increased homelessness, crime, and suicide, partly owing to insufficient, adequate, and accessible community resources. Therefore, appropriate resource placement is a key deinstitutionalization issue. The study's aim was to identify residential group homes for people with mental illness in Kochi Prefecture, Japan, and the social resources necessary for social reintegration using a geographic information system (GIS). Everyday living areas (ELAs), as defined by the Japanese Community-Based Integrated Care System for People with Mental Illness (CICSM), were assessed using ELA location simulations. We used GIS to determine the spatial distribution of group homes, visiting nursing stations, psychiatric hospitals, daycare centers, and employment support offices. Following the CICSM definition of ELAs, we identified areas that people with mental illness could reach within 30 min on foot/by bicycle and counted the number of social resources in them. The ELA location simulation results suggest that policymakers should avoid uniform distribution of ELAs according to the CICSM definition. Establishing ELAs in suburban areas requires careful consideration of the available community resources, number of people with mental illness, existing support systems, and feasibility of the location.

**Keywords:** mental illness; community-based integrated care system for people with mental illness; everyday living area; group home; GIS





## 1. Introduction

In developed countries, the deinstitutionalization of psychiatric care began in the 1950s [1] through the closure of psychiatric hospitals and reductions in the number of psychiatric beds [2]. A London-based assessment of UK deinstitutionalization policy reported the successful conversion of psychiatric hospitals into local, small, and community-based residential care facilities [3]. However, some reports indicate that deinstitutionalization increases both mortality and suicide rates in previous inhabitants of psychiatric hospitals [4]. Some studies also suggest that homelessness and incarceration have increased in this population [1,5,6]. In the United States, incarceration rates of people with mental illness have been increasing [7]. These patterns indicate that successful deinstitutionalization involves more than simply changing the location of care. Rather, service plans should be tailored to individual needs and people with mental illness should be involved in service plan development and implementation. The biggest challenge to achieving this goal is the creation of appropriate and accessible community resources [8].

It has long been pointed out that the number of hospital beds for psychiatric patients and the long-term hospitalization of such patients are higher in Japan than in other countries [9,10]. Furthermore, in recent years, the number of patients with mental illness has increased [11]. The burden of medical costs associated with an aging psychiatric population and long-term hospitalization has become a major issue, and Japan has strengthened its efforts toward deinstitutionalization [12]. In 2017, the Japanese government put forward measures to promote community living for all members of the population, regardless of

whether or not they had a mental disorder. The community-based integrated care system (CICS) was introduced to manage the rapidly aging population in Japan. To provide comprehensive care that integrates various local resources [13], the CICS involves the collaboration of authorized health, medical, and welfare experts and includes informal and mutual activities by residents (e.g., by volunteers). Using the CICS as a reference, the government identified the need to establish a community-based integrated care system for people with mental illness (CICSM) [14,15]. Whereas the target population of the CICS is older people, that of the CICSM is people with mental illness of all ages (from children to older people) [16]. The CICSM is a framework for the comprehensive provision of medical care, disability welfare/nursing care, housing, social participation (employment), mutual help in the community, and education. Within the context of the rapidly aging population in Japan, the CICSM can help people with mental illness to live independently in a familiar community [14,15].

The CICS emphasizes the need to develop social resources so that older people who need support can receive a wide range of services continuously. The area in which such services are provided is called an everyday living area (ELA) and is defined as an area that a health care provider can reach within 30 min, using the district surrounding a Japanese junior high school as a guideline [17,18]. Like the CICS, the CICSM defines an ELA as an area that provides a comprehensive range of services, including not only medical care and nursing care but also welfare services [19]. An ELA is an area where patients with mental illness can be discharged to their own communities after completing hospital treatment and receive services such as medical care, welfare, and housing. The ELA is determined by the local government [19,20] according to the setting of each region.

Therefore, it is assumed that ELAs contain the social resources to provide these services and that patients have free access to them [15]. It has been reported previously that community-based psychosocial and psychological interventions for psychiatric patients may reduce readmissions more than standard inpatient care [21–23]. The Japanese government established the ELA as an area within which people with mental illness can receive support for living in the community following deinstitutionalization. The aim is to provide the same amount of support that such individuals previously received in psychiatric hospitals. Long-term psychosocial and psychological interventions are important for patients with mental illness to enable them to live in the community without experiencing relapses. For example, such interventions can involve team visits to a patient's home to provide psychosocial and functional support, clinical assessment, and crisis intervention. Therefore, the existence of resources that enable patients to receive psychosocial and psychological interventions in ELAs as defined by the CICS is particularly important to ensure that patients receive treatment in the community.

Geographic information systems (GIS) are useful for visualizing local social resources and examining their spatial distribution and relationships. This technology uses geographic location to manage, process, and visually display spatial data with location-related information, enabling advanced analysis and rapid decision-making [24]. GIS contributes to public health problem solving and health care policymaking [25,26]. For example, the Centers for Disease Control and Prevention has been using GIS since the 1990s to track and map infectious disease outbreaks and to monitor and assess geographic factors and outcomes for diabetes, heart disease, and stroke prevention [27]. GIS can be used to accurately analyze and depict the distribution of the population requiring mental health care and can help to explain factors that affect the accessibility of health care services [28].

Research on social resources related to mental health has focused on accessibility to mental health services [29–31] and identifying gaps in community mental health policy and planning [32,33]. Many contextual factors associated with mental health have been reported; for example, one study identified a relationship between environmental exposure and mental health [34]. A case study from Japan that used GIS examined the validity of the ELA as defined by the CICS [35]. However, no reports have identified resources within CICSM-defined ELAs and evaluated ELAs for people with mental illness living in the

community. Taking into account the unintended consequences of deinstitutionalization for psychiatric patients in developed countries (e.g., homelessness and crime [36]), the aim of the Japanese ELA system is to geographically connect people with mental illness with community resources. The system may provide useful information for developing alternative services in other countries. In this study, we identified residential group homes (GHs) for people with mental illness in Kochi Prefecture, Japan, and the social resources necessary for the social reintegration of such individuals. Furthermore, we used GIS and an ELA location simulation to evaluate ELAs as defined by the CICSM. We hope that the study findings will contribute to ELA location and policymaking for the social reintegration of people with mental illness.

## 2. Materials and Methods

### 2.1. Target Area

As part of the Evaluation Project for Implementation of Community-Based Integrated Care System for People with Mental Illness in Kochi Prefecture, Kochi Prefecture was selected as a target area. Kochi is located in the southern half of the Shikoku region (it comprises 37.8% of this region) and faces the Pacific Ocean [37] (Figure 1). Kochi Prefecture has the highest percentage (84%) of forested land of all prefectures in Japan [38]. As of May 2023, the population was 669,287, and [37] approximately 47% of the total population of Kochi Prefecture lives in Kochi City (which has a population of 317,831) [39]. In June 2016, the number of psychiatric hospital outpatients was approximately 30,000 [40] (Figure 2).

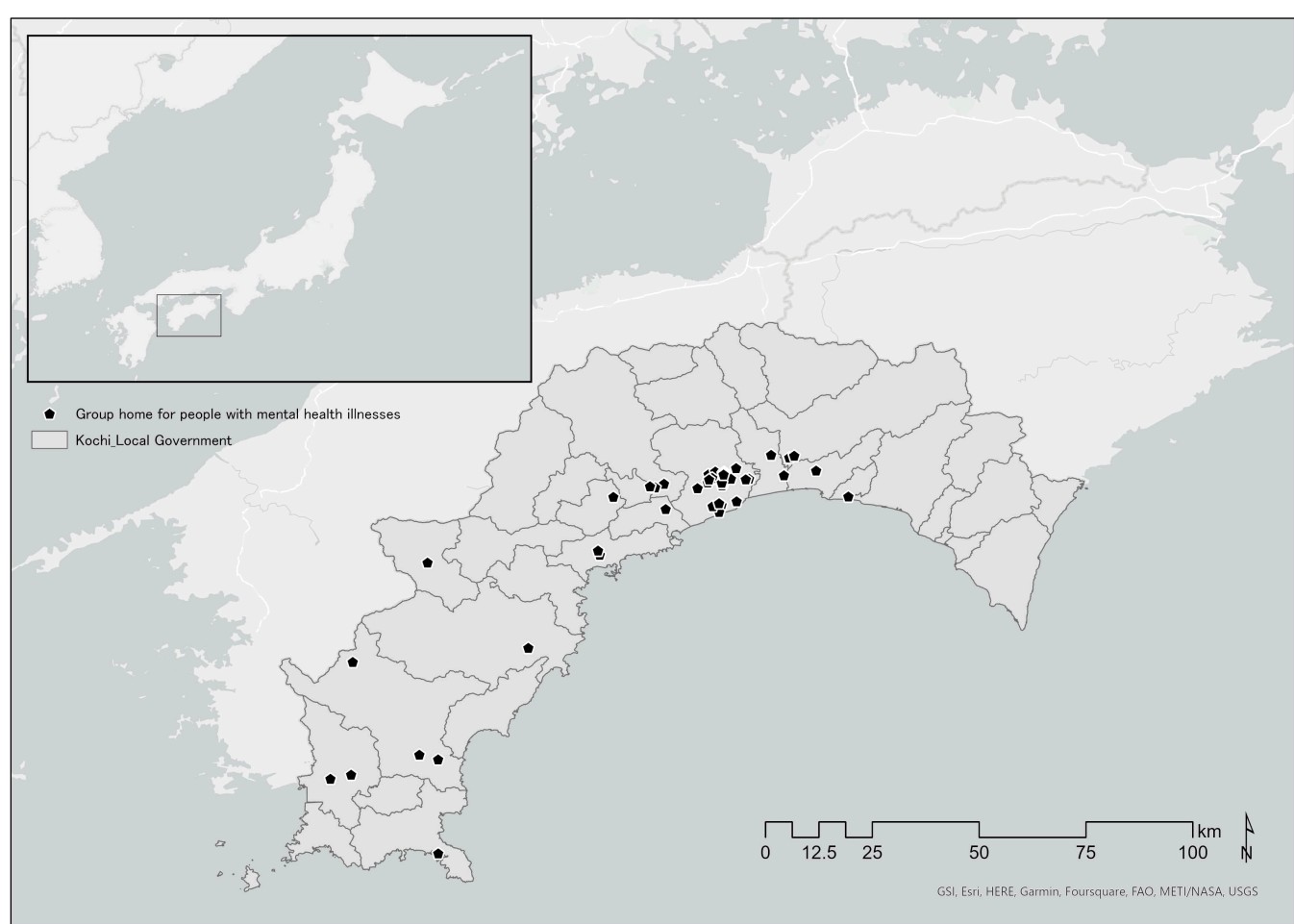

**Figure 1.** Location of Kochi Prefecture and spatial distribution of group homes.

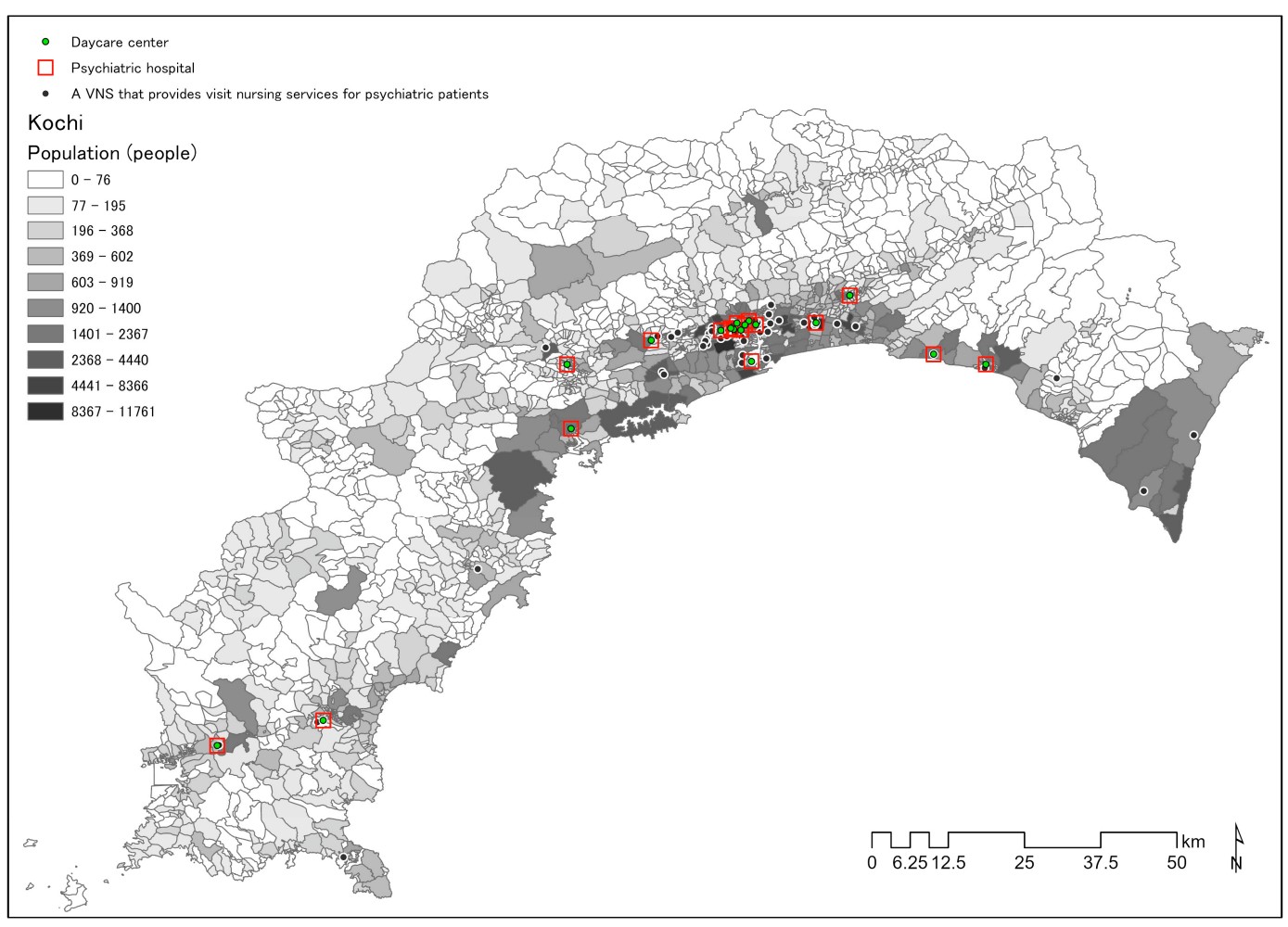

**Figure 2.** Population distribution in Kochi Prefecture and spatial distribution of daycare centers, psychiatric hospitals, and visiting nursing stations.

### 2.2. Data Collection

To investigate the spatial distribution of social resources, we used the names and addresses of the following social resources published on the Internet: GHs [41,42], psychiatric hospitals (PHs) [43,44], daycare centers (DCs), support offices for continuous employment type B (SCEBs) [41], and visiting nursing stations (VNSs) [45,46]. In addition, we investigated GH capacity, acceptance of people with mental illness, PH bed numbers and DC provision, the full-time equivalent number of VNS nurses, and provision of home-visit nursing services to psychiatric patients. Data sources, types, and uses are shown in Table 1.

### 2.3. Analysis Methods

We used open data to visualize the location of GHs inhabited by people with mental illness using GIS and to analyze ELA location characteristics to determine whether people with mental illness had access to home care services.

#### 2.3.1. Distribution of GHs, VNSs, PHs, DCs, and SCEBs

We counted GH capacity, PH beds, PHs with DCs, SCEB capacity, and VNSs that provided psychiatric nursing services. Median values (ranges) for the number of PH beds, capacity of GHs and SCEBs, and the number of nurses calculated on a full-time equivalent basis were obtained.

**Table 1.** Data sources used for analysis.

| No. | Data Type | Source | Use |
|-----|-----------|--------|-----|
| 1 | Kochi Prefecture (polygon) | National Land Numerical Information, download site [47]. | We created a polygon layer of Kochi Prefecture divided according to local government. |
| 2 | Address of group home for people with mental illness (point) | Website of Kochi Prefecture [42] and Kochi City [41]. | Using GIS, we converted the address into coordinate values and plotted the center of the building on the map. |
| 3 | Psychiatric hospital (point) | Website of Kochi Prefecture [43] and Kochi Medical Social Workers Association [44]. | |
| 4 | Support office for continuous employment type B (point) | Website of Kochi Prefecture [42] and Kochi City [41]. | |
| 5 | Visiting nursing station (point) | Website of Kochi Prefecture [45] and Kochi Prefecture Visiting Nurse Association [46]. | |

GIS, geographic information system.

### 2.3.2. ELA Location Simulation

Using GIS, the addresses of each GH, PH, DC, SCEB, and VNS were converted into coordinate values and the center of each building was plotted on the map. Taking into account the direction of travel for VNS provision and the direction of travel for people with mental illness to receive services, we performed the following three ELA location simulations:

(1)  ELA 1: An area that can be reached by a visiting nurse in 30 min by car from each VNS.
(2)  ELA 2: Areas reachable on foot by people with mental illness from each GH.
(3)  ELA 3: Areas reachable by bicycle by people with mental illness from each GH.

Transportation in (1) was defined as a car. In Japan, because of the declining birthrate and aging population, public transportation networks in local cities are declining [48]. Therefore, (2) and (3) were defined as walking and cycling with reference to studies on mobility for people with mental illness [49,50]. The travel speed in (1) was defined as the legal speed for automobiles based on the road traffic network. Walking speed in (2) was defined as 5 km/h, which is the default walk time in ArcGIS Pro. The moving speed of the bicycle in (3) was set at 11 km/h, with reference to a report by Morota et al. [51] and the minutes of a meeting of the Ministry of Land, Infrastructure, Transport and Tourism [52]. Using the Service Area Analysis tool in ArcGIS Pro, we created a polygon layer of the service areas defined in (1), (2), and (3) and depicted them on the map. A polygon layer represents the area of interest and depicts the results of spatial data analysis on a base map. The color of the polygon can be changed to represent the results of an operation (e.g., changing flooded areas to blue).

### 2.3.3. PH, DC, and SCEB Capacities in ELAs

To identify the GHs outside the home-visit nursing service area of each VNS, we visually confirmed those GHs outside the polygon layer that were 30 min distance from the VNS as calculated by (1). Next, the points of PHs, DCs, and SCEBs in the polygon layer that was 30 min by foot and by bicycle from GHs as determined in (2) and (3) were counted using the Summarize Within tool (Table S1).

### 3. Results

*3.1. Overview of GHs, PHs, DCs, SCEBs, and VNSs in Kochi Prefecture*

In Kochi Prefecture, we identified 73 GHs, 111 SCEBs, and 41 PHs, 18 of which have DCs. There were 53 VNSs providing visiting nursing services for people with mental illness (Table 2).

**Table 2.** Characteristics of group homes, psychiatric hospitals, daycare centers, support offices for continuous employment type B, and visiting nurse stations.

| Characteristics | Median (Range) | n | % |
|---|---|---|---|
| Group home | | 73 | 100 |
| Group home admission capacity (people) | 12.0 (4–99) | | |
| Accepts people with mental illness | | 49 | 67.1 |
| Psychiatric hospital | | 41 | 100 |
| Psychiatric hospital number of beds (beds) | 60.0 (0–271) | | |
| Daycare center * | | 18 | 43.9 |
| Support office for continuous employment type B | | 111 | 100 |
| Support office for continuous employment type B admission capacity (people) | 20.0 (10–44) | | |
| Visiting nursing station | | 80 | 100 |
| Visiting nursing station number of nurses calculated on a full-time equivalent basis (people) | 4.5 (2.5–16.5) | | |
| Visiting nursing station providing visiting nursing care services to psychiatric patients | | 53 | 66.2 |

* A daycare center for psychiatric patients attached to a psychiatric hospital.

### 3.2. Results of ELA Location Simulation

#### 3.2.1. ELA 1: GH Location in an Area Reachable by Car from a VNS in 30 Min

The simulation of ELA 1 and the visual confirmation of Figure 3 showed that the two GHs indicated by arrows were outside the scope of home nursing care service delivery (Figure 3).

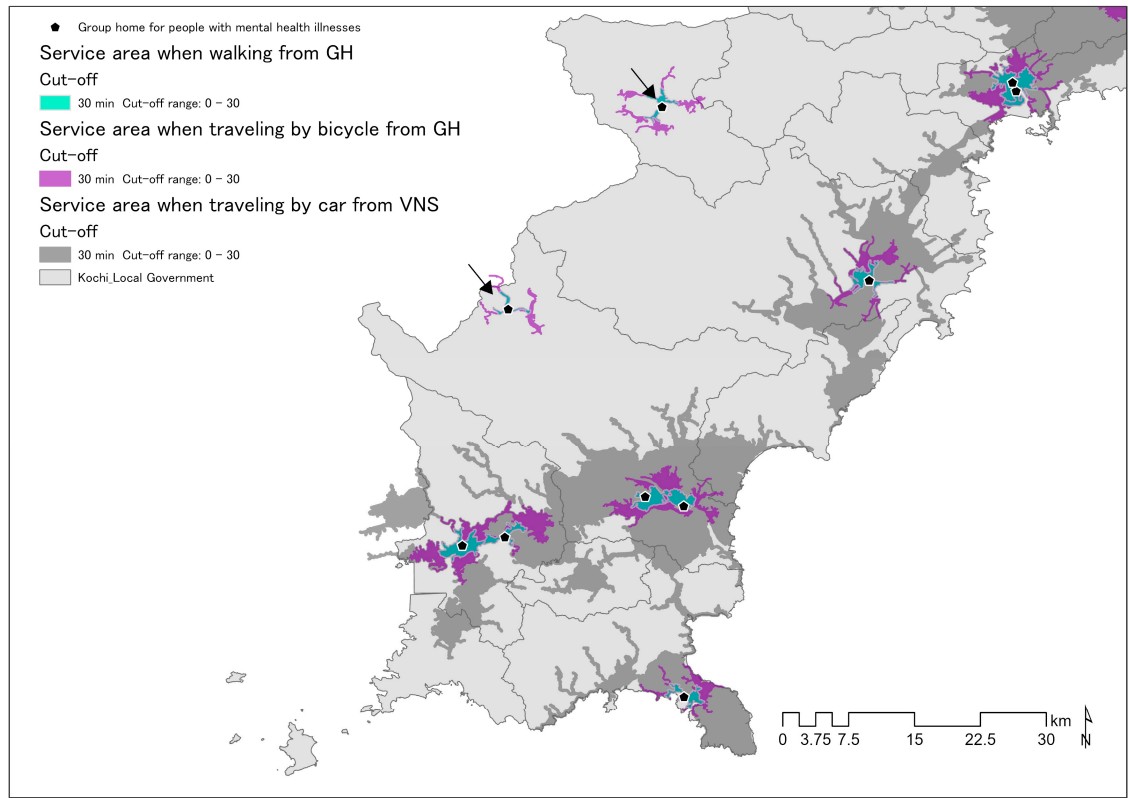

**Figure 3.** Location of GHs outside ELAs of VNSs. GH, group home; ELA, everyday living area; VNS, visiting nursing station.

### 3.2.2. ELA 2, 3: Location of Social Resources within the ELA When Walking and Cycling from GH

The social resources within the ELA were calculated, assuming that each GH resident's mode of transportation was walking or cycling.

Table S1 shows the summary results of the analysis using the Summarize Within tool using GIS. The social resources within each ELA were as follows. ELA IDs 6 and 11 had 0 social resources (Figure 4, Table S1). ELA ID 2 had 0 social resources accessible by foot. ELA IDs 3, 12, 18, and 22 had SCEBs in the ELA, but no PHs or DCs. Of ELAs, 36.7% accessible by foot had no PHs. Of ELAs accessible by bicycle, 14.3% had no PHs. Table S1 shows the aggregated results of the analysis of resources within each ELA.

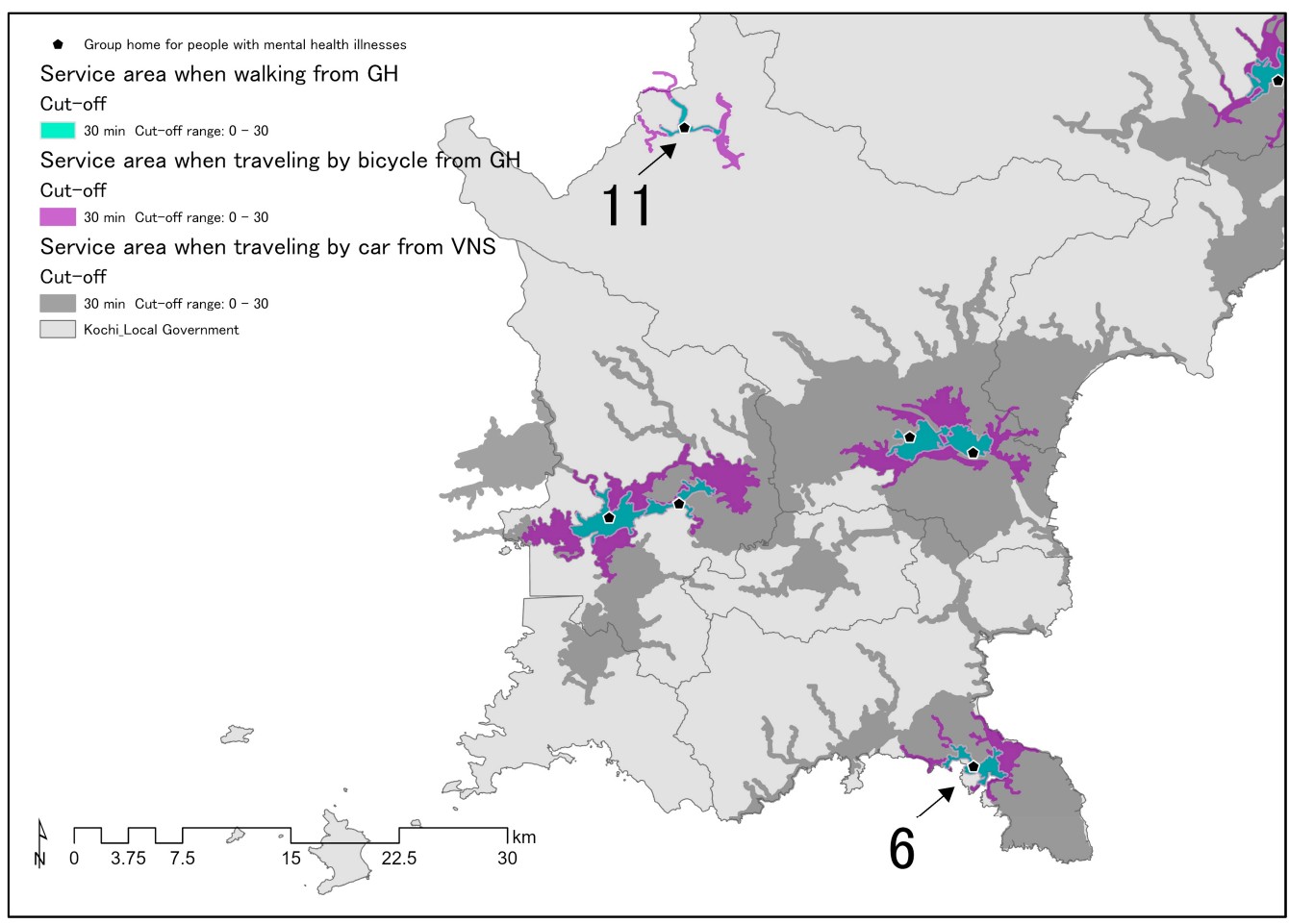

**Figure 4.** GHs with no social resources within the ELA if traveling on foot or by bicycle. GH, group home; ELA, everyday living area; VNS, visiting nursing station.

The ELA location simulation results suggest that policymakers should avoid the uniform distribution of ELAs according to the CICSM definition. Establishing ELAs in suburban areas requires careful consideration of the available community resources, the number of people with mental illness, existing support systems and the feasibility of the location.

### 4. Discussion

Using GIS, an ELA location simulation was performed with the target area as Kochi Prefecture. As mentioned above, we used the CICSM definition of ELAs as areas providing a wide range of medical, nursing, and welfare services.

Considering the direction of movement of visiting nurses, if ELAs are located in an area that can be reached by car from VNSs in 30 min, people in GHs with mental illness

within ELA IDs 6 and 11 are not eligible for home-visit nursing services. Home-visit nursing interventions have been shown to prevent the readmission of psychiatric patients and thereby help them to live in the community [53]. In Japan, it has been reported that there is a need for VNS-based living support for psychiatric patients and for the provision of information about available social resources [54]. The provision of home-visit nursing is not mandatory for ELAs as defined by the CICSM; however, considering the need for recurrence prevention for people with mental illness, the need for support in daily living, and the need for support such as medication adherence, the lack of VNSs within an ELA may be a barrier in determining a regional ELA. Therefore, the area served by a VNS may be an important factor for policymakers to consider when deciding on an ELA.

GHs in Japan are so-called community-based residential rehabilitation units and are responsible for improving the practical life skills of former long-term psychiatric inpatients and helping them to transfer to the community. A community care unit model of psychiatric rehabilitation has demonstrated positive trends in improved mental health and social functioning, reduced involuntary treatment, and readmission rates [55,56]. Social ties improve mental health in general, not just in people with mental illness [57]. It is particularly important for people with mental illness to be able to connect with PH services and receive ongoing home support. This is because such links can help in the early identification of mental health problems in people with mental illness and prevent re-hospitalization. VNSs, DCs, and SCEBs are responsible for continuous support for people with mental illness living in the community, so these resources are essential to enable such individuals to remain in the community. People with mental illness in ELA IDs 2, 6, and 11 were not able to access PHs, DCCs, and SCEBs when ELAs were located in areas that could be reached on foot within 30 min of GHs. Additionally, for ELAs in areas reachable by bicycle within 30 min from GHs, people with mental illness in ELA IDs 6 and 11 were unable to access any resources. Lack of aftercare contact with a psychiatric facility and four or more hospitalizations have been cited as predictors of readmission in psychiatric patients after discharge [58]. The inability to walk or cycle to a hospital or DC can be a barrier to regular visits and ongoing daycare support. During psychiatric rehabilitation, access to social resources and the availability, or lack of alternative services are factors that influence service utilization for people with mental illness [55]. Kochi has the highest ratio of mountainous areas by prefecture, at 89% [59]. Considering the economic rationale for generating self-supporting businesses such as GHs, it is understandable that GH proliferate in central areas with large populations. However, policymakers should avoid the uniform distribution of ELA according to the CICSM definition. A strategy is needed to allocate resources that takes into account alternative services and access methods to resources, particularly in suburban areas where social resources are scarce. To implement such a strategy, it may be necessary to compare the costs of hospitalization for people with mental illness with alternatives to hospitalization that involve community support. In many countries, suburban areas with small populations have fewer community resources than larger urban areas. A large-scale study in France found that increased availability of community-based resources and nursing home capacity for persons with disabilities were associated with reduced involuntary admission rates. Therefore, we suggest that a geographically uniform distribution of mental health professionals to meet the needs of the population is important to prevent involuntary admission to hospitals [60]. GIS-based simulations of hypothetical ELAs could help policymakers to redistribute community resources in a geographically uniform manner. Decisions about the location of ELAs in Kochi Prefecture should use GIS to consider the population, social resources, and geographical characteristics of each individual ELA location. In this study, we performed simulations using open data. However, additional studies are needed to investigate in detail the living conditions of people with mental illness and the support status of specialists.

This study had several limitations. We used open data published on the Internet and did not consider the existing status of service provision. The VNS service area shown in this simulation depicts the area that can be reached by road in 30 min in each direction,

according to the CICSM ELA definition. However, VNSs may actually provide home nursing services to areas that are more than 30 min away in each direction. The findings of this simulation could inform community resource development to help people with mental illness to continue living in the community. However, this simulation was for one region in Japan, so may not be generalizable to other countries.

## 5. Conclusions

The easiest option for policymakers choosing the siting of ELAs in Kochi Prefecture is to use the CICSM definition for densely populated areas. However, ELA decisions in sparsely populated suburbs must consider access to social resources and alternative services. Given that professional support for the continuation of treatment is essential for people with mental illness when making ELA decisions, policymakers should consider social resources that allow people with mental illness to connect with others and receive support. We recommend that policymakers avoid uniformly distributing ELAs according to the CICSM definition. It is recommended that decisions about the feasibility of suburban ELAs should include careful examination of the actual community resources, the number of people with mental illness, and the existing support systems. Considering the important role of VNSs for people with mental illness in Japan, it may be necessary to implement a system to enable people with mental illness to receive home-visit nursing services or to access an alternative service. Community service solutions that use CICSM-defined ELAs in Japan could contribute to solving similar problems in countries that provide small-scale community-based residential care. We recommend ELA simulation using GIS service area analysis to confirm the optimum siting of ELAs according to the specific situation of particular regions.

**Supplementary Materials:** The following supporting information can be downloaded at: https://www.mdpi.com/article/10.3390/challe14030030/s1, Table S1: Number of PHs, DCs, and SCEBs by ELA obtained using simulation.

**Author Contributions:** Conceptualization, Y.N. and H.N.; methodology, Y.N. and H.N., software, Y.N. and H.N.; validation, Y.N. and H.N.; formal analysis, Y.N. and H.N.; investigation, Y.N.; resources, Y.N.; data curation, Y.N. and H.N.; writing—original draft preparation, Y.N. and H.N.; writing—review and editing, Y.N. and H.N.; visualization, Y.N. and H.N.; supervision, H.N.; project administration, H.N.; funding acquisition, Y.N. All authors have read and agreed to the published version of the manuscript.

**Funding:** This research received no external funding.

**Institutional Review Board Statement:** Ethical review and approval were waived for this study because open data available online were used. The data do not include any personal information.

**Informed Consent Statement:** Not applicable.

**Data Availability Statement:** Publicly available datasets were analyzed in this study.

**Acknowledgments:** We thank Diane Williams for editing a draft of this manuscript.

**Conflicts of Interest:** The authors declare no conflict of interest.

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
