# Peer review of "Evaluation of Everyday Living Areas for Deinstitutionalized Community-Living People with Mental Illness"

_challenges, doi:10.3390/challe14030030_

Round 1

Reviewer 1 Report

The manuscript is interestingbut confusing and to much short to be understandable. I just have a few comments that I hope will improve it:

- l.18: 30 minutes walking/biking can be too much for other countries so it is important to state why this cut-off is adequate for Japan

- l.44: present some figures or cite work that justify this sentence

- l.78: since this paper will be read by people not familiar wih GIS, it would be better to start by a definition

- l.81: also the distribution of the population needing mental health care

- l.101: it does not make sense this section. This should be integrated on introduction

- l.130: the explanation is not enough to clarify what has been done

- l.167: there should be a map with the location of the place of residence and aother one with the population distribution and the location of services

- l.170: not clear what the range is about

-l.186: writen in japonese

-l-198: it is not clear whow the ELAs where defined

Reviewer 2 Report

The topic of manuscript (consequences of deistituzionalization and service's accessiblity) is of great interes. Authors presente a valid reesearch work with a smart and interesting design.

My only suggestion is to shorten introduction section and review Tabel S1 (not very redable/useful in present form)

Reviewer 3 Report

This is a short study of the distribution of some health, employment and social care services within an 'Everyday Living Area' (ELA) as defined by the Japanese governmental model of a Community Based Integrated Care System for people with Mental Illness (CICSM).

The study does not include any innovation methodologically in terms of GIS mapping, data organisation or form of analysis, the principal originality is in the subject matter of providing some form of evaluation of CICSM based ELA's, within a particular Japanese Prefecture. (Kochi). This evaluation is provided soley on the time taken to access key services through three principal forms of movement. I think the study will be of interest to health professionals and spatial planners and policy makers both within Japan and beyond.

The paper is clearly written and succinct with enough relevant citations around its subject but with virtually no background research highlighted on different forms of GIS mapping as part of the methodology design. I think this comes through in the rudimentary form of the maps presented which are limited in their ability to communicate findings. I would question the single scale of map utilised particularly. 

My feeling is that there should be an overall map illustrating each GH and accompanying ELA outline within the prefecture. It appears from table S1 that there are 49 in total. I would also like to see an example of more detailed mapping of GH, VNS, PH, DC and SCEB within a particular prefecture for a more detailed understanding of distribution and accessibility within a prefecture. It should also I think include a more detailed representation of routes and forms of travel. I believe this would add some important further depth to the study and more originality in the form of GIS mapping utilised. This would also allow for a more detailed evaluation to occur across more than one ELA in relation to distribution and accessibility which would add more originality.

Reviewer 4 Report

The paper addresses the use of geographic information system methods to ascertain the availability of community-based services for people with mental illness.  The paper is enormously interesting.  The research has been carefully conducted and should be very useful to a broad readership. There are a few things that could be easily done to improve this paper and specially to make the work more accessible to a non-technical audience.  The introduction is primarily focussed on de-institutionalisation, but the reader does struggle to get a good sense about the extent to which GIS methods have been previously used in analogue applications.  It would be useful to understand the reason for choosing the Kochi prefecture. Is it because this prefecture has the relevant data to perform the methods? I found it difficult to determine the actual aim of the work. Is it to determine the feasibility and usefulness of the methods or is to report on an evaluation of the prefecture using the methods? The two things seem to be reported on in an interchangeable way. Some of the terms are difficult to understand. Notably there are three ELA location simulations, but these do not seem to be linked to the ELA IDs.  What is a polygon? The essence of the findings are really hard to grasp.  There seems to be two ELA IDs that have fewer social services. Is this right? What are we to make of this? Two IDs out of how many? Overall, what am I to conclude about the level of services in the prefecture? This is what I don’t understand because the discussion seems to circle back to the rationale in the introduction about the need for social services. I don’t think this point needs to be remade. It is already made earlier on. 

Round 2

Reviewer 3 Report

I would outline the extents of figure 3 on figure 2 for orientation. otherwise happy to recommend for approval.

Final grammar and spell-check required.